# Pareto Multi-Task Learning

**Xi Lin[1], Hui-Ling Zhen[1], Zhenhua Li[2], Qingfu Zhang[1], Sam Kwong[1]**

[1]City University of Hong Kong, [2]Nanjing University of Aeronautics and Astronautics

`xi.lin@my.cityu.edu.hk`, `huilzhen@um.cityu.edu.hk`, `zhenhua.li@nuaa.edu.cn`
`{qingfu.zhang, cssamk}@cityu.edu.hk`

## Abstract

Multi-task learning is a powerful method for solving multiple correlated tasks simultaneously. However, it is often impossible to find one single solution to optimize all the tasks, since different tasks might conflict with each other. Recently, a novel method is proposed to find one single Pareto optimal solution with good trade-off among different tasks by casting multi-task learning as multiobjective optimization. In this paper, we generalize this idea and propose a novel Pareto multi-task learning algorithm (Pareto MTL) to find a set of well-distributed Pareto solutions which can represent different trade-offs among different tasks. The proposed algorithm first formulates a multi-task learning problem as a multiobjective optimization problem, and then decomposes the multiobjective optimization problem into a set of constrained subproblems with different trade-off preferences. By solving these subproblems in parallel, Pareto MTL can find a set of well-representative Pareto optimal solutions with different trade-off among all tasks. Practitioners can easily select their preferred solution from these Pareto solutions, or use different trade-off solutions for different situations. Experimental results confirm that the proposed algorithm can generate well-representative solutions and outperform some state-of-the-art algorithms on many multi-task learning applications.

## 1 Introduction

Multi-task learning (MTL) [1], which aims at learning multiple correlated tasks at the same time, is a popular research topic in the machine learning community. By solving multiple related tasks together, MTL can further improve the performance of each task and reduce the inference time for conducting all the tasks in many real-world applications. Many MTL approaches have been proposed in the past, and they achieve great performances in many areas such as computer vision [2], natural language processing [3] and speech recognition [4].

Most MTL approaches are proposed for finding one single solution to improve the overall performance of all tasks [5, 6]. However, it is observed in many applications that some tasks could conflict with each other, and no single optimal solution can optimize the performance of all tasks at the same time [7]. In real-world applications, MTL practitioners have to make a trade-off among different tasks, such as in self-driving car [8], AI assistance [9] and network architecture search [10, 11].

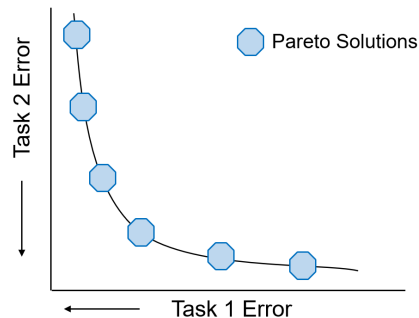

Figure 1: **Pareto MTL** can find a set of widely distributed Pareto solutions with different trade-offs for a given MTL. Then the practitioners can easily select their preferred solution(s).

How to combine different tasks together and make a proper trade-off among them is a difficult problem. In many MTL applications, especially those using deep multi-task neural networks, all tasks are first combined into a single surrogate task via linear weighted scalarization. A set of fixed weights, which reflects the practitioners' preference, is assigned to these tasks. Then the single surrogate task is optimized. Setting proper weights for different tasks is not easy and usually requires exhaustive weights search. In fact, no single solution can achieve the best performance on all tasks at the same time if some tasks conflict with each other.

Recently, Sener and Koltun [12] formulate a multi-task learning problem as a multi-objective optimization problem in a novel way. They propose an efficient algorithm to find one Pareto optimal solution among different tasks for a MTL problem. However, the MTL problem can have many (even an infinite number of ) optimal trade-offs among its tasks, and the single solution obtained by this method might not always satisfy the MTL practitioners' needs.

In this paper, we generalize the multi-objective optimization idea [12] and propose a novel Pareto Multi-Task Learning (Pareto MTL) algorithm to generate a set of well-representative Pareto solutions for a given MTL problem. As shown in Fig. 1, MTL practitioners can easily select their preferred solution(s) among the set of obtained Pareto optimal solutions with different trade-offs, rather than exhaustively searching for a set of proper weights for all tasks.

The main contributions of this paper are: [1]

- We propose a novel method to decompose a MTL problem into multiple subproblems with different preferences. By solving these subproblems in parallel, we can obtain a set of well-distributed Pareto optimal solutions with different trade-offs for the original MTL.
- We show that the proposed Pareto MTL can be reformulated as a linear scalarization approach to solve MTL with dynamically adaptive weights. We also propose a scalable optimization algorithm to solve all constrained subproblems with different preferences.
- Experimental results confirm that the proposed Pareto MTL algorithm can successfully find a set of well representative solutions for different MTL applications.

## 2 Related Work

Multi-task learning (MTL) algorithms aim at improving the performance of multiple related tasks by learning them at the same time. These algorithms often construct shared parameter representation to combine multiple tasks. They have been applied in many machine learning areas. However, most MTL algorithms mainly focus on constructing shared representation rather than making trade-offs among multiple tasks [5, 6].

Linear tasks scalarization, together with grid search or random search of the weight vectors, is the current default practice when a MTL practitioner wants to obtain a set of different trade-off solutions. This approach is straightforward but could be extremely inefficient. Some recent works [7, 13] show that a single run of an algorithm with well-designed weight adaption can outperform the random search approach with more than one hundred runs. These adaptive weight methods focus on balancing all tasks during the optimization process and are not suitable for finding different trade-off solutions.

Multi-objective optimization [14] aims at finding a set of Pareto solutions with different trade-offs rather than one single solution. It has been used in many machine learning applications such as reinforcement learning [15], Bayesian optimization [16, 17, 18] and neural architecture search [10, 19]. In these applications, the gradient information is usually not available. Population-based and gradient-free multi-objective evolutionary algorithms [20, 21] are popular methods to find a set of well-distributed Pareto solutions in a single run. However, it can not be used for solving large scale and gradient-based MTL problems.

Multi-objective gradient descent [22, 23, 24] is an efficient approach for multi-objective optimization when gradient information is available. Sener and Koltun [12] proposed a novel method for solving MTL by treating it as multi-objective optimization. However, similar to the adaptive weight methods, this method tries to balance different tasks during the optimization process and does not have a systematic way to incorporate trade-off preference. In this paper, we generalize it for finding a set of well-representative Pareto solutions with different trade-offs among tasks for MTL problems.

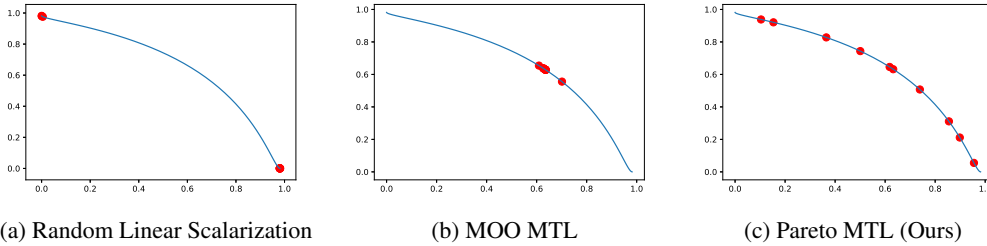

(a) Random Linear Scalarization    (b) MOO MTL    (c) Pareto MTL (Ours)

Figure 2: The convergence behaviors of different algorithms on a synthetic example. (a) The obtained solutions of random linear scalarization after 100 runs. (b) The obtained solutions of the MOO-MTL [12] method after 10 runs. (c) The obtained solutions of the Pareto MTL method proposed by this paper after 10 runs. The proposed Pareto MTL successfully generates a set of widely distributed Pareto solutions with different trade-offs. Details of the synthetic example can be found in section 5.

# 3 Multi-Task Learning as Multi-Objective Optimization

## 3.1 MTL as Multi-Objective Optimization

A MTL problem involves a set of $m$ correlated tasks with a loss vector:

$$\min_\theta \mathcal{L}(\theta) = (\mathcal{L}_1(\theta), \mathcal{L}_2(\theta), \cdots, \mathcal{L}_m(\theta))^\mathrm{T}, \tag{1}$$

where $\mathcal{L}_i(\theta)$ is the loss of the $i$-th task. A MTL algorithm is to optimize all tasks simultaneously by exploiting the shared structure and information among them.

Problem (1) is a multi-objective optimization problem. No single solution can optimize all objectives at the same time. What we can obtain instead is a set of so-called Pareto optimal solutions, which provides different optimal trade-offs among all objectives. We have the following definitions [25]:

**Pareto dominance.** Let $\theta^a, \theta^b$ be two points in $\Omega$, $\theta^a$ is said to dominate $\theta^b$ ($\theta^a \prec \theta^b$) if and only if $\mathcal{L}_i(\theta^a) \leq \mathcal{L}_i(\theta^b), \forall i \in \{1, ..., m\}$ and $\mathcal{L}_j(\theta^a) < \mathcal{L}_j(\theta^b), \exists j \in \{1, ..., m\}$.

**Pareto optimality.** $\theta^*$ is a Pareto optimal point and $\mathcal{L}(\theta^*)$ is a Pareto optimal objective vector if it does not exist $\hat{\theta} \in \Omega$ such that $\hat{\theta} \prec \theta^*$. The set of all Pareto optimal points is called the Pareto set. The image of the Pareto set in the loss space is called the Pareto front.

In this paper, we focus on finding a set of well-representative Pareto solutions that can approximate the Pareto front. This idea and the comparison results of our proposed method with two others are presented in Fig. 2.

## 3.2 Linear Scalarization

Linear scalarization is the most commonly-used approach for solving multi-task learning problems. This approach uses a linear weighted sum method to combine the losses of all tasks into a single surrogate loss:

$$\min_\theta \mathcal{L}(\theta) = \sum_{i=1}^m \boldsymbol{w}_i \mathcal{L}_i(\theta), \tag{2}$$

where $\boldsymbol{w}_i$ is the weight for the $i$-th task. This approach is simple and straightforward, but it has some drawbacks from both multi-task learning and multi-objective optimization perspectives.

In a typical multi-task learning application, the weights $\boldsymbol{w}_i$ are needed to be assigned manually before optimization, and the overall performance is highly dependent on the assigned weights. Choosing a proper weight vector could be very difficult even for an experienced MTL practitioner who has expertise on the given problem.

Solving a set of linear scalarization problems with different weight assignments is also not a good idea for multi-objective optimization. As pointed out in [26, Chapter 4.7], this method can only provide solutions on the convex part of the Pareto front. The linear scalarization method with different weight assignments is unable to handle a concave Pareto front as shown in Fig. 2.

### 3.3 Gradient-based method for multi-objective optimization

Many gradient-based methods have been proposed for solving multi-objective optimization problems [22, 23]. Fliege and Svaiter [24] have proposed a simple gradient-based method, which is a generalization of a single objective steepest descent algorithm. The update rule of the algorithm is $\theta_{t+1} = \theta_t + \eta d_t$ where $\eta$ is the step size and the search direction $d_t$ is obtained as follows:

$$(d_t, \alpha_t) = \arg \min_{d \in R^n, \alpha \in R} \alpha + \frac{1}{2}||d||^2, s.t. \ \nabla \mathcal{L}_i(\theta_t)^T d \leq \alpha, i = 1, ..., m. \tag{3}$$

The solutions of the above problem will satisfy:

**Lemma 1 [24]:** Let $(d^k, \alpha^k)$ be the solution of problem (3).

1. If $\theta_t$ is Pareto critical, then $d_t = 0 \in \mathbb{R}^n$ and $\alpha_t = 0$.
2. If $\theta_t$ is not Pareto critical, then

$$\begin{aligned} \alpha_t &\leq -(1/2)||d_t||^2 < 0, \\ \nabla \mathcal{L}_i(\theta_t)^T d_t &\leq \alpha_t, i = 1, ..., m, \end{aligned} \tag{4}$$

where $\theta$ is called Pareto critical if no other solution in its neighborhood can have better values in all objective functions. In other words, if $d_t = \mathbf{0}$, no direction can improve the performance for all tasks at the same time. If we want to improve the performance for a specific task, another task's performance will be deteriorated (e.g., $\exists i, \mathcal{L}_i(\theta_t)^T d_t > 0$). Therefore, the current solution is a Pareto critical point. When $d_t \neq \mathbf{0}$, we have $\nabla \mathcal{L}_i(\theta_t)^T d_t < 0, i = 1, ..., m$, which means $d_t$ is a valid descent direction for all tasks. The current solution should be updated along the obtained direction $\theta_{t+1} = \theta_t + \eta d_t$.

Recently, Sener and Koltun [12] used the multiple gradient descent algorithm (MGDA) [22] for solving MTL problems and achieve promising results. However, this method does not have a systemic way to incorporate different trade-off preference information. As shown in Fig. 2, running the algorithm multiple times can only generate some solutions in the middle of the Pareto front on the synthetic example. In this paper, we generalize this method and propose a novel Pareto MTL algorithm to find a set of well-distributed Pareto solutions with different trade-offs among all tasks.

## 4 Pareto Multi-Task Learning

### 4.1 MTL Decomposition

We propose the Pareto Multi-Task Learning (Pareto MTL) algorithm in this section. The main idea of Pareto MTL is to decompose a MTL problem into several constrained multi-objective subproblems with different trade-off preferences among the tasks in the original MTL. By solving these subproblems in parallel, a MTL practitioner can obtain a set of well-representative solutions with different trade-offs.

Decomposition-based multi-objective evolutionary algorithm [27, 28], which decomposes a multi-objective optimization problem (MOP) into several subproblems and solves them simultaneously, is one of the most popular gradient-free multi-objective optimization methods. Our proposed Pareto MTL algorithm generalizes the decomposition idea for solving large-scale and gradient-based MTL.

We adopt the idea from [29] and decompose the MTL into $K$ subproblems with a set of well-distributed unit preference vectors $\{\boldsymbol{u}_1, \boldsymbol{u}_2, ..., \boldsymbol{u}_K\}$ in $R_+^m$. Suppose all objectives in the MOP are non-negative, the multi-objective subproblem corresponding to the preference vector $\boldsymbol{u}_k$ is:

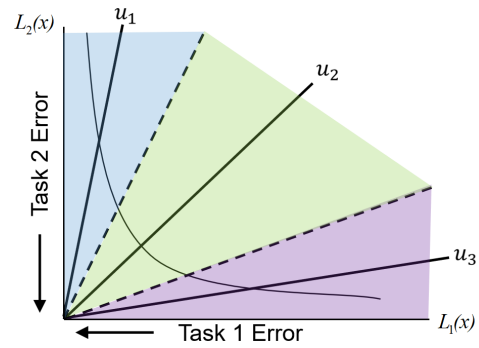

Figure 3: **Pareto MTL** decomposes a given MTL problem into several subproblems with a set of preference vectors. Each MTL subproblem aims at finding one Pareto solution in its restricted preference region.

$$\min_{\theta} \mathcal{L}(\theta) = (\mathcal{L}_1(\theta), \mathcal{L}_2(\theta), \cdots, \mathcal{L}_m(\theta))^{\mathrm{T}}, s.t. \quad \mathcal{L}(\theta) \in \Omega_k, \tag{5}$$

where $\Omega_k(k = 1, ..., K)$ is a subregion in the objective space:

$$\Omega_k = \{\boldsymbol{v} \in R_+^m | \boldsymbol{u}_j^T \boldsymbol{v} \le \boldsymbol{u}_k^T \boldsymbol{v}, \forall j = 1, ..., K\} \tag{6}$$

and $\boldsymbol{u}_j^T \boldsymbol{v}$ is the inner product between the preference vector $\boldsymbol{u}_j$ and a given vector $\boldsymbol{v}$. That is to say, $\boldsymbol{v} \in \Omega_k$ if and only if $\boldsymbol{v}$ has the smallest acute angle to $\boldsymbol{u}_k$ and hence the largest inner product $\boldsymbol{u}_k^T \boldsymbol{v}$ among all $K$ preference vectors.

The subproblem (5) can be further reformulated as:

$$\min_{\theta} \mathcal{L}(\theta) = (\mathcal{L}_1(\theta), \mathcal{L}_2(\theta), \cdots, \mathcal{L}_m(\theta))^{\mathrm{T}}$$
$$s.t. \quad \mathcal{G}_j(\theta_t) = (\boldsymbol{u}_j - \boldsymbol{u}_k)^T \mathcal{L}(\theta_t) \le 0, \forall j = 1, ..., K, \tag{7}$$

As shown in Fig. 3, the preference vectors divide the objective space into different sub-regions. The solution for each subproblem would be attracted by the corresponding preference vector and hence be guided to its representative sub-region. The set of solutions for all subproblems would be in different sub-regions and represent different trade-offs among the tasks.

## 4.2 Gradient-based Method for Solving Subproblems

### 4.2.1 Finding the Initial Solution

To solve the constrained multi-objective subproblem (5) with a gradient-based method, we need to find an initial solution which is feasible or at least satisfies most constraints. For a randomly generated solution $\theta_r$, one straightforward method is to find a feasible initial solution $\theta_0$ which satisfies:

$$\min_{\theta_0} ||\theta_0 - \theta_r||^2 \quad s.t. \quad \mathcal{L}(\theta_0) \in \Omega_k. \tag{8}$$

However, this projection approach is an $n$ dimensional constrained optimization problem [30]. It is inefficient to solve this problem directly, especially for a deep neural network with millions of parameters. In the proposed Pareto MTL algorithm, we reformulate this problem as unconstrained optimization, and use a sequential gradient-based method to find the initial solution $\theta_0$.

For a solution $\theta_r$, we define the index set of all activated constraints as $I(\theta_r) = \{j | \mathcal{G}_j(\theta_r) \ge 0, j = 1, ..., K\}$. We can find a valid descent direction $d_r$ to reduce the value of all activated constraints $\{\mathcal{G}_j(\theta_r) | j \in I(\theta_r)\}$ by solving:

$$(d_r, \alpha_r) = \arg \min_{d \in R^n, \alpha \in R} \alpha + \frac{1}{2}||d||^2, s.t. \nabla \mathcal{G}_j(\theta_r)^T d \le \alpha, j \in I(\theta_r). \tag{9}$$

This approach is similar to the unconstrained gradient-based method (3), but it reduces the value of all activated constraints. The gradient-based update rule is $\theta_{r_{t+1}} = \theta_{r_t} + \eta_r d_{r_t}$ and will be stopped once a feasible solution is found or a predefined number of iterations is met.

### 4.2.2 Solving the Subproblem

Once we have an initial solution, we can use a gradient-based method to solve the constrained subproblem. For a constrained multiobjective optimization problem, the Pareto optimality restricted on the feasible region $\Omega_k$ can be defined as [24]:

**Restricted Pareto Optimality.** $\theta^*$ is a Pareto optimal point for $\mathcal{L}(\theta)$ restricted on $\Omega_k$ if $\theta^* \in \Omega_k$ and it does not exist $\hat{\theta} \in \Omega_k$ such that $\hat{\theta} \prec \theta^*$.

According to [24, 30], we can find a descent direction for this constrained MOP by solving a subproblem similar to the subproblem (3) for the unconstrained case:

$$(d_t, \alpha_t) = \arg \min_{d \in R^n, \alpha \in R} \alpha + \frac{1}{2}||d||^2$$
$$s.t. \quad \nabla \mathcal{L}_i(\theta_t)^T d \le \alpha, i = 1, ..., m. \tag{10}$$
$$\nabla \mathcal{G}_j(\theta_t)^T d \le \alpha, j \in I_\epsilon(\theta_t),$$

where $I_\epsilon(\theta)$ is the index set of all activated constraints:

$$I_\epsilon(\theta) = \{j \in I | \mathcal{G}_j(\theta) \geq -\epsilon\}. \tag{11}$$

We add a small threshold $\epsilon$ to deal with the solutions near the constraint boundary. Similar to the unconstrained case, for a feasible solution $\theta_t$, by solving problem (10), we either obtain $d_t = \mathbf{0}$ and confirm that $\theta_t$ is a Pareto critical point restricted on $\Omega_k$, or obtain $d_t \neq \mathbf{0}$ as a descent direction for the constrained multi-objective problem (7). In the latter case, if all constraints are inactivated (e.g., $I_\epsilon(\theta) = \emptyset$), $d_t$ is a valid descent direction for all tasks. Otherwise, $d_t$ is a valid direction to reduce the values for all tasks and all activated constraints.

**Lemma 2 [30]:** Let $(d^k, \alpha^k)$ be the solution of problem (10).

1. If $\theta_t$ is Pareto critical restricted on $\Omega_k$, then $d_t = \mathbf{0} \in \mathbb{R}^n$ and $\alpha_t = 0$.
2. If $\theta_t$ is not Pareto critical restricted on $\Omega_k$, then

$$
\begin{aligned}
&\alpha_t \leq -(1/2)||d_t||^2 < 0, \\
&\nabla \mathcal{L}_i(\theta_t)^T d_t \leq \alpha_t, i = 1, ..., m \\
&\nabla \mathcal{G}_j(\theta_t)^T d_t \leq \alpha_t, j \in I_\epsilon(\theta_t).
\end{aligned} \tag{12}
$$

Therefore, we can obtain a restricted Pareto critical solution for each subproblem with simple iterative gradient-based update rule $\theta_{t+1} = \theta_t + \eta_r d_t$. By solving all subproblems, we can obtain a set of diverse Pareto critical solutions restricted on different sub-regions, which can represent different trade-offs among all tasks for the original MTL problem.

### 4.2.3 Scalable Optimization Method

By solving the constrained optimization problem (10), we can obtain a valid descent direction for each multi-objective constrained subproblem. However, the optimization problem itself is not scalable well for high dimensional decision space. For example, when training a deep neural network, we often have more than millions of parameters to be optimized, and solving the constrained optimization problem (10) in this scale would be extremely slow. In this subsection, we propose a scalable optimization method to solve the constrained optimization problem.

Inspired by [24], we first rewrite the optimization problem (10) in its dual form. Based on the KKT conditions, we have

$$d_t = -(\sum_{i=1}^m \lambda_i \nabla \mathcal{L}_i(\theta_t) + \sum_{j \in I_\epsilon(x)} \beta_i \nabla \mathcal{G}_j(\theta_t)), \quad \sum_{i=1}^m \lambda_i + \sum_{j \in I_\epsilon(\theta)} \beta_j = 1, \tag{13}$$

where $\lambda_i \geq 0$ and $\beta_i \geq 0$ are the Lagrange multipliers for the linear inequality constraints. Therefore, the dual problem is:

$$
\begin{aligned}
&\max_{\lambda_i, \beta_j} \quad -\frac{1}{2}||\sum_{i=1}^m \lambda_i \nabla \mathcal{L}_i(\theta_t) + \sum_{j \in I_\epsilon(x)} \beta_i \nabla \mathcal{G}_j(\theta_t)||^2 \\
&s.t. \quad \sum_{i=1}^m \lambda_i + \sum_{j \in I_\epsilon(\theta)} \beta_j = 1, \lambda_i \geq 0, \beta_j \geq 0, \forall i = 1, ..., m, \forall j \in I_\epsilon(\theta).
\end{aligned} \tag{14}
$$

For the above problem, the decision space is no longer the parameter space, and it becomes the objective and constraint space. For a multiobjective optimization problem with 2 objective function and 5 activated constraints, the dimension of problem (14) is 7, which is significantly smaller than the dimension of problem (10) which could be more than a million.

The algorithm framework of Pareto MTL is shown in **Algorithm** 1. All subproblems can be solved in parallel since there is no communication between them during the optimization process. The only preference information for each subproblem is the set of preference vectors. Without any prior knowledge for the MTL problem, a set of evenly distributed unit preference vectors would be a reasonable default choice, such as $K + 1$ preference vectors $\{(cos(\frac{k\pi}{2K}), sin(\frac{k\pi}{2K}))|k = 0, 1, ..., K\}$ for 2 tasks. We provide more discussion on preference vector setting and sensitivity analysis of the preference vectors in the supplementary material.

---
**Algorithm 1** Pareto MTL Algorithm
---
1: **Input:** A set of evenly distributed vectors $\{\boldsymbol{u}_1, \boldsymbol{u}_2, ..., \boldsymbol{u}_K\}$
2: **Update Rule:**
3: (can be solved in parallel)
4: **for** $k = 1$ to $K$ **do**
5:      randomly generate parameters $\theta_r^{(k)}$
6:      find the initial parameters $\theta_0^{(k)}$ from $\theta_r^{(k)}$ using gradient-based method
7:      **for** $t = 1$ to $T$ **do**
8:         obtain $\lambda_{ti}^{(k)} \geq 0, \beta_{ti}^{(k)} \geq 0, \forall i = 1, ..., m, \forall j \in I_\epsilon(\theta)$ by solving subproblem (14)
9:         calculate the direction $d_t^{(k)} = -(\sum_{i=1}^m \lambda_{ti}^{(k)} \nabla \mathcal{L}_i(\theta_t^{(k)})) + \sum_{j \in I_{\epsilon(\boldsymbol{x})}} \beta_{ti}^{(k)} \nabla \mathcal{G}_j(\theta_t^{(k)})$
10:         update the parameters $\theta_{t+1}^{(k)} = \theta_t^{(k)} + \eta d_t^{(k)}$
11:      **end for**
12: **end for**
13: **Output:** The set of solutions for all subproblems with different trade-offs $\{\theta_T^{(k)} | k = 1, \cdot, K\}$
---

## 4.3 Pareto MTL as an Adaptive Linear Scalarization Approach

We have proposed the Pareto MTL algorithm from the multi-objective optimization perspective. In this subsection, we show that the Pareto MTL algorithm can be reformulated as a linear scalarization of tasks with adaptive weight assignment. In this way, we can have a deeper understanding of the differences between Pareto MTL and other MTL algorithms.

We first tackle the unconstrained case. Suppose we do not decompose the multi-objective problem and hence remove all constraints from the problem (14), it will immediately reduce to the update rule proposed by MGDA [22] which is used in [12]. It is straightforward to rewrite the corresponding MTL into a linear scalarization form:

$$\mathcal{L}(\theta_t) = \sum_{i=1}^m \lambda_i \mathcal{L}_i(\theta_t), \tag{15}$$

where we adaptively assign the weights $\lambda_i$ by solving the following problem in each iteration:

$$\max_{\lambda_i} -\frac{1}{2} \| \sum_{i=1}^m \lambda_i \nabla \mathcal{L}_i(\theta_t) \|^2, \quad s.t. \sum_{i=1}^m \lambda_i = 1, \quad \lambda_i \geq 0, \forall i = 1, ..., m. \tag{16}$$

In the constrained case, we have extra constraint terms $\mathcal{G}_j(\theta_t)$. If $\mathcal{G}_j(\theta_t)$ is inactivated, we can ignore it. For an activated $\mathcal{G}_j(\theta_t)$, assuming the corresponding reference vector is $\boldsymbol{u}_k$, we have:

$$\nabla \mathcal{G}_j(\theta_t) = (\boldsymbol{u}_j - \boldsymbol{u}_k)^T \nabla \mathcal{L}(\theta_t) = \sum_{i=1}^m (\boldsymbol{u}_{ji} - \boldsymbol{u}_{ki}) \nabla \mathcal{L}_i(\theta_t). \tag{17}$$

Since the gradient direction $d_t$ can be written as a linear combination of all $\nabla \mathcal{L}_i(\theta_t)$ and $\nabla \mathcal{G}_j(\theta_t)$ as in (13), the general Pareto MTL algorithm can be rewritten as:

$$\mathcal{L}(\theta_t) = \sum_{i=1}^m \alpha_i \mathcal{L}_i(\theta_t), \text{ where } \alpha_i = \lambda_i + \sum_{j \in I_{\epsilon(\theta)}} \beta_j (\boldsymbol{u}_{ji} - \boldsymbol{u}_{ki}), \tag{18}$$

where $\lambda_i$ and $\beta_j$ are obtained by solving problem (14) with assigned reference vector $\boldsymbol{u}_k$.

Therefore, although MOO-MTL [12] and Pareto MTL are both derived from multi-objective optimization, they can also be treated as linear MTL scalarization with adaptive weight assignments. Both methods are orthogonal to many existing MTL approaches. We provide further discussion on the adaptive weight vectors in the supplementary material.

## 5 A Synthetic Example

To better analyze the convergence behavior of the proposed Pareto MTL, we first compare it with two commonly used methods, namely the linear scalarization method and the multiple gradient descent

algorithm used in MOO-MTL [12], on a simple synthetic multi-objective optimization problem:

$$\min_{\boldsymbol{x}} f_1(\boldsymbol{x}) = 1 - \exp\left(-\sum_{i=1}^{d}(\boldsymbol{x}_d - \frac{1}{\sqrt{d}})^2\right)$$

$$\min_{\boldsymbol{x}} f_2(\boldsymbol{x}) = 1 - \exp\left(-\sum_{i=1}^{d}(\boldsymbol{x}_d + \frac{1}{\sqrt{d}})^2\right)$$

(19)

where $f_1(\boldsymbol{x})$ and $f_2(\boldsymbol{x})$ are two objective functions to be minimized at the same time and $\boldsymbol{x} = (\boldsymbol{x}_1, \boldsymbol{x}_2, ..., \boldsymbol{x}_d)$ is the $d$ dimensional decision variable. This problem has a concave Pareto front on the objective space.

The results obtained by different algorithms are shown in Fig. 2. In this case, the proposed Pareto MTL can successfully find a set of well-distributed Pareto solutions with different trade-offs. Since MOO-MTL tries to balance different tasks during the optimization process, it gets a set of solutions with similar trade-offs in the middle of the Pareto front in multiple runs. It is also interesting to observe that the linear scalarization method can only generate extreme solutions for the concave Pareto front evenly with 100 runs. This observation is consistent with the theoretical analysis in [26] that the linear scalarization method will miss all concave parts of a Pareto front. It is evident that fixed linear scalarization is not always a good idea for solving the MTL problem from the multi-objective optimization perspective.

# 6  Experiments

In this section, we compare our proposed Pareto MTL algorithm on different MTL problems with the following algorithms: 1) **Single Task**: the single task baseline; 2) **Grid Search**: linear scalarization with fixed weights; 3) **GradNorm** [13]: gradient normalization; 4) **Uncertainty** [7]: adaptive weight assignments with uncertainty balance; 5) **MOO-MTL** [12]: finding one Pareto optimal solution for multi-objective optimization problem. More experimental results and discussion can be found in the supplementary material.

## 6.1  Multi-Fashion-MNIST

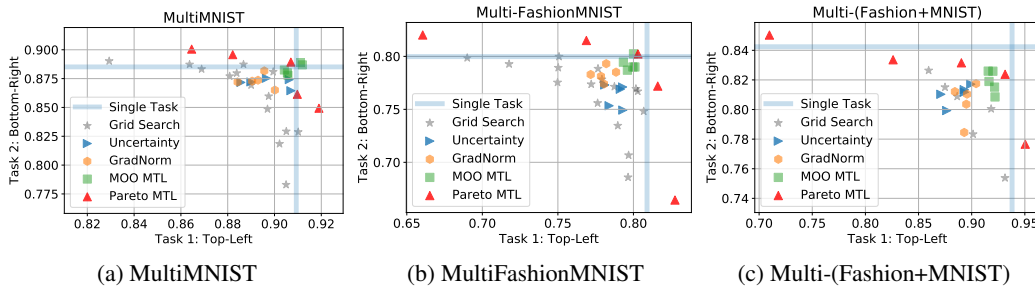

(a) MultiMNIST          (b) MultiFashionMNIST          (c) Multi-(Fashion+MNIST)

Figure 4: **The results for three experiments with Task1&2 accuracy:** our proposed Pareto MTL can successfully find a set of well-distributed solutions with different trade-offs for all experiments, and it significantly outperforms Grid Search, Uncertainty and GradNorm. MOO-MTL algorithm can also find promising solutions, but their diversity is worse than the solutions generated by Pareto MTL.

In order to evaluate the performance of Pareto MTL on multi-task learning problems with different tasks relations, we first conduct experiments on MultiMNIST [31] and two MultiMNIST-like datasets. To construct the MultiMNIST dataset, we randomly pick two images with different digits from the original MNIST dataset [32], and then combine these two images into a new one by putting one digit on the top-left corner and the other one on the bottom-right corner. Each digit can be moved up to 4 pixels on each direction. With the same approach, we can construct a MultiFashionMINST dataset with overlap FashionMNIST items [33], and a Multi-(Fashion + MNIST) with overlap MNIST and FashionMNIST items. For each dataset, we have a two objective MTL problem to classify the item

on the top-left (task 1) and to classify the item on the bottom-right (task 2). We build a LeNet [32] based MTL neural network similar to the one used in [12]. The obtained results are shown in Fig. 4.

In all experiments, since the tasks conflict with each other, solving each task separately results in a hard-to-beat single-task baseline. Our proposed Pareto MTL algorithm can generate multiple well-distributed Pareto solutions for all experiments, which are compatible with the strong single-task baseline but with different trade-offs among the tasks. Pareto MTL algorithm achieves the overall best performance among all MTL algorithms. These results confirm that our proposed Pareto MTL can successfully provide a set of well-representative Pareto solutions for a MTL problem.

It is not surprising to observe that the Pareto MTL's solution for subproblems with extreme preference vectors (e.g., $(0, 1)$ and $(1, 0)$) always have the best performance in the corresponding task. Especially in the Multi-(Fashion-MNIST) experiment, where the two tasks are less correlated with each other. In this problem, almost all MTL's solutions are dominated by the strong single task's baseline. However, Pareto MTL can still generate solutions with the best performance for each task separately. It behaves as auxiliary learning, where the task with the assigned preference vector is the main task, and the others are auxiliary tasks.

Pareto MTL uses neural networks with simple hard parameter sharing architectures as the base model for MTL problems. It will be very interesting to generalize Pareto MTL to other advanced soft parameter sharing architectures [5]. Some recently proposed works on task relation learning [34, 35, 36] could also be useful for Pareto MTL to make better trade-offs for less relevant tasks.

## 6.2 Self-Driving Car: Localization

| Method | Reference Vector | Translation (m) | Rotation (°) |
|---|---|---|---|
| Single Task | - | 8.392 | 2.461 |
| Grid Search | (0.25,0.75) | 9.927 | 2.177 |
| | (0.5,0.5) | 7.840 | 2.306 |
| | (0.75,0.25) | 7.585 | 2.621 |
| GradNorm | - | 7.751 | 2.287 |
| Uncertainty | - | 7.624 | 2.263 |
| MOO-MTL | - | 7.909 | 2.090 |
| Pareto MTL | (0,1) | **7.285** | 2.335 |
| | $(\frac{\sqrt{2}}{2}, \frac{\sqrt{2}}{2})$ | 7.724 | 2.156 |
| | (1,0) | 8.411 | **1.771** |

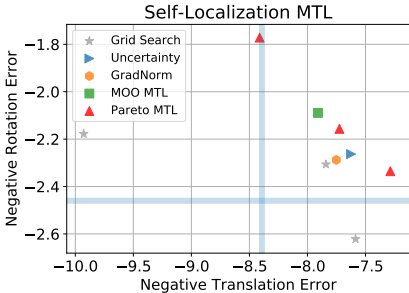

Figure 5: **The results of self-location MTL experiment:** Our proposed Pareto MTL outperforms other algorithms and provides solutions with different trade-offs.

We further evaluate Pareto MTL on an autonomous driving self-localization problem [8]. In this experiment, we simultaneously estimate the location and orientation of a camera put on a driving car based on the images it takes. We use data from the apolloscape autonomous driving dataset [37, 38], and focus on the Zpark sample subset. We build a PoseNet with a ResNet18 [39] encoder as the MTL model. The experiment results are shown in Fig. 5. It is obvious that our proposed Pareto MTL can generate solutions with different trade-offs and outperform other MTL approaches.

We provide more experiment results and analysis on finding the initial solution, Pareto MTL with many tasks, and other relative discussions in the supplementary material.

## 7 Conclusion

In this paper, we proposed a novel Pareto Multi-Task Learning (Pareto MTL) algorithm to generate a set of well-distributed Pareto solutions with different trade-offs among tasks for a given multi-task learning (MTL) problem. MTL practitioners can then easily select their preferred solutions among these Pareto solutions. Experimental results confirm that our proposed algorithm outperforms some state-of-the-art MTL algorithms and can successfully find a set of well-representative solutions for different MTL applications.

**Acknowledgments**

This work was supported by the Natural Science Foundation of China under Grant 61876163 and Grant 61672443, ANR/RGC Joint Research Scheme sponsored by the Research Grants Council of the Hong Kong Special Administrative Region, China and France National Research Agency (Project No. A-CityU101/16), and Hong Kong RGC General Research Funds under Grant 9042489 (CityU 11206317) and Grant 9042322 (CityU 11200116).

## Footnotes

[1] The code is available at: `https://github.com/Xi-L/ParetoMTL`

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
