[Supplementary Material]

# Supplementary Material: Pareto MTL

**Xi Lin[1], Hui-Ling Zhen[1], Zhenhua Li[2], Qingfu Zhang[1], Sam Kwong[1]**
[1]City University of Hong Kong, [2]Nanjing University of Aeronautics and Astronautics
xi.lin@my.cityu.edu.hk, huilzhen@um.cityu.edu.hk, zhenhua.li@nuaa.edu.cn
{qingfu.zhang, cssamk}@cityu.edu.hk

In this supplementary material, we provide more detailed discussions and experimental results on Pareto MTL. We also point out some limitations for the current Pareto MTL and propose some potential research directions.

## 1 The Importance of Finding the Initial Solution

| (a) MOO MTL | (b) Pareto MTL w/o Initialization | (c) Pareto MTL w/ Initialization |

Figure 1: **The convergence behaviours of different algorithms on the synthetic example.** The proposed Pareto MTL algorithm without initialization step can still generate a set of solutions with different trade-offs on the Pareto front. However, it fails to find solutions near the end points.

Finding an initial solution is important for solving Pareto MTL. In this paper, we propose a gradient-based method to find an initial solution to each MTL problem. The initial solution should be a feasible solution to the constrained subproblem or at least satisfy most constraints. In this section, we further analyze the importance of finding the initial solution.

We first test the Pareto MTL algorithm without the initialization step on the synthetic example. As shown in Figure. 1, without the initialization step, Pareto MTL can still generate a set of Pareto solutions with different trade-offs and outperforms the MOO-MTL algorithm. However, the diversity of these solutions is worse than those generated by the Pareto MTL with the initialization step. It is obvious that Pareto MTL without initialization fails to find solutions near the endpoints of the Pareto front. The lack of ability to cross the boundary between different sub-regions would be one reason for inferior performance.

For a subproblem, a randomly generated solution might not be in (or even far away from) its preference sub-region. With the initialization step, the solution can be sequentially updated to get close to the assigned preference sub-region since the constraint values are lowered at each iteration. Many constraints would turn inactivated once the solution crosses its boundary to get close to its assigned preference vector. In contrast, without the initialization step, the solution might stop at some boundary of the preference vectors (with corresponding activated constraints) during the optimization process since now the objective functions are taken into consideration. It is easier to find a descent direction to lower the value of activated constraints than to find a direction to lower both the values of activated constraints and the tasks.

Figure 2: **Pareto MTL with and without Initialization on Multi-Fashion MNIST problem.**

We also test the Pareto MTL without initialization step on the Multi-FashionMNIST problem. As shown in Fig. 2, the Pareto MTL without initialization step can generate solutions with different trade-offs, but the diversity is much worse than Pareto MTL with initialization step.

In the proposed Pareto MTL, we obtain the descent direction for each subproblem by solving:

$$
\begin{aligned}
(d_t, \alpha_t) = &\arg \min_{d \in R^n, \alpha \in R} \alpha + \frac{1}{2}||d||^2 \\
&s.t. \quad \nabla \mathcal{L}_i(\theta_t)^T d \leq \alpha, i = 1, ..., m. \\
&\qquad \nabla \mathcal{G}_j(\theta_t)^T d \leq \alpha, j \in I_\epsilon(\theta_t).
\end{aligned}
\tag{1}
$$

Actually, depended on how to obtain the gradient direction, we have three different algorithms:

1. Only consider the tasks $\mathcal{L}_i(\theta_t)$: MOO-MTL, for convergence;

2. Only consider the constraints $\mathcal{G}_j(\theta_t)$: the initialization step in Pareto MTL, for diversity;

3. Consider both $\mathcal{L}_i(\theta_t)$ and $\mathcal{G}_j(\theta_t)$: the main step of Pareto MTL, tries to find a set of restricted Pareto points on diverse sub-region, somehow balance the convergence and diversity.

How to choose or switch among these different algorithms would be an interesting research topic. The proposed Pareto MTL first runs step 2 and then runs step 3. An immediate extension is to keep and get a snapshot [1] of all solutions at the end of step 3, then relax all constraints and run step 1. In this way, we can obtain a set of restricted Pareto critical solutions for each subproblem by Pareto MTL (running step 2 and step 3) with good diversity, plus a set of Pareto critical solutions (not restricted) with potential better convergence by running step 1 at the end.

## 2 MTL with Many Tasks

Pareto MTL uses a set of preference vectors to decompose a given MTL problem into several constrained multi-objective subproblems. By solving all subproblems, Pareto MTL can obtain a set of optimal solutions with different trade-offs among all tasks. However, to fairly cover the whole objective space of all tasks, the number of required preference vectors would increase exponentially when the MTL problem has more tasks.

To be concrete, Pareto MTL needs to solve a $(m + K - 1)$-dimensional constrained optimization problem to find the descent direction at each iteration, and solve $K$ subproblems in total, where $m$ is the number of tasks and $K$ is the number of preference vectors. Under mild assumption, the Pareto front would be a $m - 1$ dimensional manifold for a multi-objective optimization problem [2]. Suppose we need $p$ (e.g., 5) widely distributed Pareto solutions to properly represent different optimal trade-offs on one dimension of the Pareto front, the total required solutions could be $p^{m-1}$ (e.g., 25 solutions for three tasks and 125 solutions for four tasks) to cover the whole Pareto front. Since we need to assign one preference vector for each expected solution, the dimension of the constrained optimization problem at each iteration would be $m+p^{m-1}-1$, and there are $p^{m-1}$ MTL subproblems to be solved in total. The current Pareto MTL suffers the curse of dimensionality to cover the whole

Pareto front for a MTL with many tasks. In other words, it would be extremely time-consuming for Pareto MTL to provide a set of widely distributed solutions to explore the whole objective space for a MTL problem with many tasks.

To check the required number of solutions, we test Pareto MTL on a multi-task learning problem with three prediction tasks on the UCI census-income dataset [3, 4]. This dataset is a subset of the 1994 American Census dataset and contains $299,285$ adults' demographic information records with $40$ different features. Similar to the setting in [5], we set the income, education level, and marital status as three binary targets to be predicted:

- Task 1: whether the person's income exceeds $50K/year.
- Task 2: whether the person's education level is at least college.
- Task 3: whether the person is never married.

We build a multi-task neural network with hard parameter sharing for three tasks as the prediction model. We first convert all discrete categorical features into one-hot vectors and obtain a $482$ dimensional input feature vector for each record. The model has two hidden fully connected hidden layers with $1024$ and $128$ hidden units, and each task has its own output layer.

(a) Pareto MTL with 25 and 5 preference vectors    (b) Pareto MTL v.s. Random Search

Figure 3: Pareto MTL with different number of preference vectors and linear scalarization with random search on census dataset.

The experiment's results are shown in Fig. 3. We first compare the performance of Pareto MTL with 5 and 25 randomly generated preference vectors. From sub-figure (a), it is clear that Pareto MTL with 25 preference vectors can represent the different trade-offs among three tasks much better. MTL practitioners can easily select their preferred solutions from the obtained results. Pareto MTL with 5 preference vectors can also provide solutions with different trade-offs, but the only 5 obtain solutions can not properly represent different optimal trade-offs on the Pareto front. Therefore, a large number of preference vectors (and hence corresponding MTL subproblems) is needed to obtain a set of well-representative trade-offs for a MTL problem with more tasks. We also compare Pareto MTL with the linear scalarization method with random weight research. As shown in sub-figure (b), Pareto MTL's solutions dominate nearly all random search's solutions, which means Pareto MTL has a much better performance on this MTL problem.

In addition to the performance, how to provide information to the practitioner for making decisions is another critical issue for many tasks. Visualizing all solutions with different trade-offs for three tasks is not as clear as for two tasks, and visualization would be much more difficult for more than three tasks. We make a discussion on some potential methods for many tasks in the rest of this section.

**Finding representative solutions with preferred trade-offs.**  When the preference vectors are fixed and cannot be adaptively adjusted, MTL practitioners can still directly use Pareto MTL to generate different Pareto solutions with their preferred trade-offs for MTL problem with many tasks. As discussed in the experiment section, in Pareto MTL, the subproblem with extreme preference vector (e.g., $(0,1)$ and $(1,0)$) can be explained as auxiliary multi-task learning, where the corresponding

task is the preferred main task and the others are auxiliary tasks. Similar explains can also be applied for subproblem with a specific preference vector. In other words, once the MTL practitioners have their preferred trade-off(s) among the tasks, they can directly run Pareto MTL to find diverse Pareto solution(s) corresponding to different preference vectors. If the MTL practitioners do not have any preferred trade-off yet, they can at least run Pareto MTL with a few different preference vectors to obtain a set of diverse Pareto solutions. They can directly choose their preferred solutions or use them to summarize preferred trade-off(s) for another run.

In this section, we run Pareto MTL with only a few preference vectors for solving a MTL with three different tasks. The dataset we use is the UTKFace dataset [6], and the MTL problem is to predict human's gender, race, and age based on one image of their faces. We build a deep MTL network with Resnet18 as the encoder and a task-specific fully connected layer for each task.

Table 1: The gender accuracy, race accuracy, and age L1-loss obtained by different algorithms. The best results are highlighted. Pareto MTL can find widely distributed solutions with diverse trade-offs.

| Method | Reference Vector | Gender (Accuracy) | Race (Accuracy) | Age (L1-Loss) |
|---|---|---|---|---|
| Single Task | - | 0.879157 | 0.781265 | 13.55133 |
| Fixed MTL | $(\frac{1}{3}, \frac{1}{3}, \frac{1}{3})$ | 0.873982 | 0.783819 | 13.90006 |
| GradNorm | - | 0.880913 | 0.766011 | 13.79205 |
| Uncertainty | - | 0.879657 | 0.770406 | 13.96869 |
| MOO-MTL | - | 0.878013 | 0.778436 | 13.80779 |
| Pareto MTL | $(\frac{\sqrt{3}}{3}, \frac{\sqrt{3}}{3}, \frac{\sqrt{3}}{3})$ | 0.878049 | 0.786747 | 13.91812 |
| | $(0, \frac{\sqrt{2}}{2}, \frac{\sqrt{2}}{2})$ | **0.885552** | 0.754095 | 14.41414 |
| | $(\frac{\sqrt{2}}{2}, 0, \frac{\sqrt{2}}{2})$ | 0.872242 | **0.792814** | 14.32156 |
| | $(\frac{\sqrt{2}}{2}, \frac{\sqrt{2}}{2}, 0)$ | 0.866895 | 0.762251 | **13.51771** |

(a) gender v.s. race    (b) gender v.s. age    (c) race v.s. age

Figure 4: The 2-D projections for the results obtained by different algorithms. We report the negative age L1-loss for the sake of consistency. Pareto MTL can provide solutions with diverse trade-offs.

The experiment result is shown in Table.1 and Fig.4. It confirms that Pareto MTL with a few specific preference vectors can still find representative Pareto solutions for the practitioner's preferred trade-offs.

**Learning to find preferred Pareto solution(s).** One advantage of Pareto MTL over MOO-MTL is its ability to incorporate preference information even with only a single run (solving one subproblem, but still need a set of preference vectors to divide the objective space). Recently, some learning-based methods have been proposed to solve MTL problems [7, 8]. It is possible to propose learning-based Pareto MTL for dynamically adjusting the preference vectors to incorporate the MTL practitioner's preference, and to guide the solutions search to their preferred smaller subspace for a MTL problem.

**Methods from the multi-objective optimization community.** By formulating the MTL with many tasks as a multi-objective optimization problem, we get a many-objective optimization problem, which is indeed a popular research topic in the multi-objective optimization community [9, 10, 11, 12]. How to adopt the techniques proposed from the multi-objective optimization community to solve the MTL problem with many tasks is a potential research direction.

# 3 The Adaptive Weight Vectors

Figure 5: **The adaptive weight vectors for different algorithms during the training process for MultiMNIST experiment.** Pareto MTL behaves differently with different reference vectors. The other algorithms with adaptive weights assignment try to balance different tasks.

In the paper, we show that MOO-MTL and the proposed Pareto MTL algorithm can be reformulated as a linear scalarization of different tasks with adaptive weights assignment where the Pareto MTL algorithm can be rewritten as:

$$\mathcal{L}(\theta_t) = \sum_{i=1}^{m} \alpha_i \mathcal{L}_i(\theta_t), \text{ where } \alpha_i = \lambda_i + \sum_{j \in I_{\epsilon(\theta)}} \beta_j(\boldsymbol{u}_{ji} - \boldsymbol{u}_{ki}), \tag{2}$$

In this section, we compare the adaptive weight vectors for different algorithms during the training process. As shown in Fig. 5, Pareto MTL has clearly different weight adaption strategies for subproblems with different preference vectors, while MOO-MTL and GradNorm always try to balance different tasks.

From the view point of linear scalarization with adaptive weights, the surrogate loss for MOO-MTL, GradNorm and Uncertainty can be written as $\mathcal{L}(\theta_t) = \sum_{i=1}^{m} \lambda_i \mathcal{L}_i(\theta_t)$. Different methods have their own strategy to adapt the weight $\lambda_i$ to balance the loss function $\mathcal{L}_i(\theta_t)$. For Pareto MTL, the weight vector now has the form $\alpha_i = \lambda_i + \sum_{j \in I_{\epsilon(\theta)}} \beta_j(\boldsymbol{u}_{ji} - \boldsymbol{u}_{ki})$. While the parameter $\lambda_i$ is still for balancing different tasks, the preference term $\sum_{j \in I_{\epsilon(\theta)}} \beta_j(\boldsymbol{u}_{ji} - \boldsymbol{u}_{ki})$ will guide the Pareto MTL to its corresponding preference vector. As shown in Fig. 5, Pareto MTL will bias the search to a specific task when it has extreme preference vectors (e.g., $(0, 1)$ and $(1, 0)$), and it will try to balance different tasks with a balance preference vector (e.g., $(\sqrt{2}/2, \sqrt{2}/2)$).

When all constraints are inactivated (e.g., $I_{\epsilon(\theta)} = \emptyset$), Pareto MTL has a feasible solution right in the assigned sub-region for a given subproblem. In this case, the surrogate loss could be reduced to $\mathcal{L}_i(\theta_t) = \sum_{i=1}^{m} \lambda_i \mathcal{L}_i(\theta_t)$ which is the same as MOO-MTL, and Pareto MTL will try to find a balanced solution in the assigned sub-region.

Pareto MTL is not mutually exclusive with other adaptive weight strategies such as GradNorm [13] and Uncertainty [14], especially when it can be reformulated as the linear scalarization method. For MTL problem with highly unbalanced tasks with different difficulties, it is possible to first balance all tasks with some adaptive weight strategies, and then use Pareto MTL to find a set of Pareto solutions for the balanced tasks. We will discuss the Pareto MTL's performance on tasks with different difficulties in the next section.

# 4 Pareto MTL with Different Tasks Difficulties

(a) Linear ($a_1 = 2, a_2 = 1$)  (b) MOO MTL ($a_1 = 2, a_2 = 1$)  (c) Pareto MTL ($a_1 = 2, a_2 = 1$)

(d) Linear ($a_1 = 10, a_2 = 1$)  (e) MOO MTL ($a_1 = 10, a_2 = 1$)  (f) Pareto MTL ($a_1 = 10, a_2 = 1$)

(g) Linear ($a_1 = 50, a_2 = 1$)  (h) MOO MTL ($a_1 = 50, a_2 = 1$)  (i) Pareto MTL ($a_1 = 50, a_2 = 1$)

Figure 6: The convergence behaviours of different algorithms on a synthetic example with different tasks difficulties. Pareto MTL can find a set of widely distributed Pareto solutions on problems with low or medium difficulty unbalance level (($a_1 = 2, a_2 = 1$) and ($a_1 = 10, a_2 = 1$)), but its performance gets worse for problem with high difficulty unbalance level ($a_1 = 50, a_2 = 1$).

Pareto MTL implicitly assumes the tasks in a MTL problem should have similar difficulties, and uses the set of widely distributed unit vectors as the preference vectors. However, its performance might be deteriorated for tasks with extremely different difficulties. It is hard to control the tasks' difficulties in real-world MTL applications manually. To clearly demonstrate the performance of Pareto MTL, we test it on the following synthetic example:

$$
\min f_1(\boldsymbol{x}) = \alpha_1 - \alpha_1 \exp\left(-\sum_{i=1}^{d}(x_d - \frac{1}{\sqrt{d}})^2\right),
$$

$$
\min f_2(\boldsymbol{x}) = \alpha_2 - \alpha_2 \exp\left(-\sum_{i=1}^{d}(x_d + \frac{1}{\sqrt{d}})^2\right),
$$

(3)

where $f_1(\boldsymbol{x})$ and $f_2(\boldsymbol{x})$ are two objective functions to be minimized at the same time and $\boldsymbol{x} = (\boldsymbol{x}_1, \boldsymbol{x}_2, ..., \boldsymbol{x}_d)$ is the $d$ dimensional decision variable. This problem has a concave Pareto front on the objective space. The two objective function can have different difficulty levels controlled by the parameters $\alpha_i$. If $\alpha_1 = \alpha_2$, the two tasks have similar difficult level, which is the synthetic example we have in the main paper. If $\alpha_1 > \alpha_2$, the task one could be "easier" since it has a larger gradient value, and task 2 could be "easier" if $\alpha_1 < \alpha_2$. We notice the difficulty measurement for real-world multi-task learning problem would be much more complicated. Here we focus on the much simplified version and left the analysis for real-world problems in the future.

**Performance for Problems with Unbalanced Difficulties.** The results on problems with different levels of unbalanced difficulties are shown in Fig. 6. As in the balanced difficult case, the linear scalarization and MOO-MTL approach can not obtain a set of well-representative solutions for all problems. Pareto MTL can find a set of widely distributed Pareto solutions on problems with low or medium difficulty unbalance level ($(a_1 = 2, a_2 = 1)$ and $(a_1 = 10, a_2 = 1)$), but its performance gets worse for problem with high difficulty unbalance level ($a_1 = 50, a_2 = 1$).

**Different Biases for MOO-MTL and Pareto MTL.** Another interesting observation in this experiment is that MOO-MTL and Pareto MTL will be biased to different tasks in the same unbalance problem. For example, in the highly unbalanced problem ($a_1 = 50, a_2 = 1$), MOO-MTL is biased to the "easier" Task 1 since it has a much larger absolute gradient value. Most solutions found by Pareto MTL, however, are biased to the "harder" Task 2. When the tasks have different difficulty levels, the decomposed sub-region would be highly unbalanced for evenly distributed preference vectors, and the solutions would be attracted by a few preference vector much easier. In this case, most solutions are attracted by the preference vector corresponding to task 2 rather than task 1.

As discussed in the previous section, combining Pareto MTL and other adaptive weight methods to balance different tasks would be one possible method for tackling different levels of task difficulties. Learning-based self-adaptive method methods would be another important research direction.

# 5 Preference Vector Assignment

Figure 7: **Three different sets of preference vector:** (a) 5 evenly distributed unit preference vectors; (b) 8 evenly distributed preference unit vectors; and (c) 5 biased unit preference vectors.

The final distribution of the solutions obtained by Pareto MTL depends on both the preference vectors and the shape of the Pareto front. Even for tasks with similar difficulties, it is still important to properly assign the set of preference vectors for Pareto MTL. When we do not have any prior information for a given MTL problem, it is reasonable to decompose the objective space for different tasks with a set of evenly distributed preference vectors as shown in Fig. 7 (a)(b). When the MTL practitioners have their own preference for a given MTL problem, they can feel free to use a set of biased preference vectors as in Fig. 7 (c).

However, it is hard to tell whether the assigned preference vectors would be the optimal one before the actual run of Pareto MTL. We run Pareto MLT multiple times with a different set of randomly generated unit preference vectors, and show the results in Fig. 8. Pareto MTL with different sets of random preference vectors can consistently generate well-distributed solutions, but the accuracy performance would be better or worse than the default uniform setting. Getting stuck in bad local Pareto optima would be one possible reason for some inferior performance, since Pareto MTL can only guarantee locally restricted Pareto optimality. Our preliminary experiment results also show that too close preference vectors (and hence narrow region $\Omega_k$) could also lead to worse performance.

How to efficiently set the preference vectors based on the user's preference or any prior information could be an interesting topic. A strategy to adaptively set or change the preference vectors during the multi-task learning process to incorporate the practitioner's preference or better explore the objective space is another possible extension.

Figure 8: Pareto MTL with different sets of randomly generated unit preference vectors.

# 6 The Gap between Optimization and Generalization

(a) MultiMNIST: Train Loss

(b) MultiMNIST: Train Accuracy

(c) MultiMNIST: Test Accuracy

(d) MultiFashion: Train Loss

(e) MultiFashion: Train Accuracy

(f) MultiFashion: Test Accuracy

(g) Fashion&MNIST: Train Loss

(h) Fashion&MNIST: Train Acc

(i) Fashion&MNIST: Test Acc

Figure 9: The training loss/training accuracy/test accuracy of Pareto MTL and other algorithms with ResNet18 on the MultiMNIST, MultiFashionMNIST and Multi-(Fashion+MNIST) datasets. The labels legend is on the last figure (bottom-right). Pareto MTL has different patterns on the training loss and the test accuracy.

Current work shows that the adaptive gradient methods would have inferior performance on some tasks [15]. In section 4 of this supplementary material, we have discussed tasks with different difficulty levels which could be one possible reason for the inferior performance. In this section, we provide another discussion based on the gap between optimization and generalization.

Pareto MTL is derived from the view of optimization. To be concrete, we reformulate MTL as a multi-objective optimization problem, and then decompose it into multiple constrained multi-objective subproblems. We also propose an efficient algorithm to solve each constrained subproblem, and treat the obtained solutions as the Pareto solutions for the original MTL problem. However, the objective functions we truly optimize is the **training loss functions** but not the **training accuracy** or even the **test accuracy** for the original MLT problem. Therefore, there would a gap between the objective functions we truly optimize and the MTL generalization ability, which the practitioners most care about. Pareto MTL might have different performance on the training loss and the test accuracy.

To show a clear gap between optimization and generalization on purpose, we test Pareto MTL with ResNet-18 model on the three MNIST-like multi-task datasets, namely the MultiMNIST, Multi-FashionMNIST, and Multi-(Fashion+MNIST) in the main paper. The ResNet-18 model could be overparameterized for the MNIST-like dataset and it has the ability to remember all training examples (and hence has a very high training accuracy). To show the different behaviors, we train the models on the MultiMNIST dataset **with early stop** and train the models on MultiFashionMNIST and Multi-(Fashion+MNIST) **till the end**.

The experimental results are shown in Fig. 9. For the training losses on all three experiments, Pareto MTL can obtain widely distributed solutions with different trade-offs. This result is not surprising since Pareto MTL makes trade-offs and optimizes the loss functions directly. However, there are different performances on the training accuracy and testing accuracy. For the MultiMNIST dataset **with early stop**, Pareto MTL has well-distributed solutions on training loss and training accuracy, but they are outperformed by the separate single-task baseline. In contrast, Pareto MTL's solutions on test accuracy are not diverse enough, but they outperform the single-task baseline and provide different optimal trade-off. For MultiFashionMNIST and Multi-(Fashion+MNIST) dataset, we train the models **till the end**. The training losses for all solutions are close to $0$, and the training accuracy for most solutions are closed to $100\%$. For this extreme case, although Pareto MTL can still generate a set of well-representative solutions on the training loss, it has worse performance on the test accuracy. Some solutions generated by Pareto MTL can match the strong performance of separate single-task baseline on the training loss, and other solutions can provide different optimal trade-offs at the same time. However, they are all outperformed by the single-task baseline on the test accuracy. In other words, Pareto MTL is still good at optimization but has a bad generalization in these cases.

Other adaptive weight methods do not have a clear advantage over the single-task baseline and linear scalarization method with fixed weights, although they sometimes can generate good solutions that match Pareto MTL's solution with the balanced trade-off. In the view of multi-objective optimization, the adaptive weight methods (with proper weight adaption strategy) should be better than linear scalarization with fixed weight. The latter can not find any solution on the concave part of a Pareto front as proved in [16]. The gap between optimization and generalization could be an important research issue when we design MTL algorithms from the view of optimization.

Another issue for Pareto MLT is the local convergence. As pointed out in the main paper, the solution for each MTL constrained subproblem is restricted Pareto optimal. If the objective functions are all convex and the constraints are properly assigned, Pareto MTL should have a set of widely distributed Pareto solutions. However, especially for training deep neural networks, the loss function would be highly non-convex and the interaction between constrains and the optimization landscape would be complicated. In this case, the solutions for Pareto MTL might be trapped by inferior local Pareto optima. How to get rid of the poor local Pareto optima is an important extension for Pareto MTL.

# 7   Conclusion Remark

In this supplementary material, we provide more experimental results and detailed discussions on Pareto MTL. We also point out several limitations of the current Pareto MTL algorithm and propose some potential research directions. Pareto MTL is derived from the view of multi-objective optimization, and it is orthogonal to many existing MTL methods. We hope there would be further developments on Pareto MTL and its applications for different MTL problems.