[Reviews · NeurIPS 2019]

Reviewer 1



-This paper investigates a better strategy to produce a wide spread of trade-offs among multiple tasks in the multi-task learning setup (as shown in Figure 2) by introducing preference vectors. Based on multi-objective optimization with the vectors, the proposed method can achieve distributed Pareto solutions. It seems to be efficient compared to other existing MTL competitors. -However, I see the contributions of this paper is incremental and limited as the goal is not that significant and the approaches are mostly based on existing strategies such as [12, 24, 30]. In addition, the approach looks based on a single shared architecture assuming there are correlated tasks, but if tasks are less relevant, I wonder how to consider the tasks in generating Pareto solutions. -The paper is well-written and easy to follow, but there are a few unclear sentences that do not give clear answers. For example, in the lines of 87-88, why existing work cannot efficiently incorporate preference information and in the lines of 173-175 why the approach in [30] is inefficient and why the sequential gradient-based method can overcome the inefficiency? -In Algorithm 1, how to evenly distribute the vectors u_i’s? Is it achieved manually or rule-based things? -Even if the proposed approach pursues an approach for generating widely distributed Pareto solutions (rather than the accuracy itself), experimental results say that the method performs better than other competitors? Is there any analysis on this? -The first experiment is based on LeNet which is a quite old network so may not show the potential of the compared approaches for the problem. Reporting results using more advanced networks would be better. ---- The authors addressed most of my concerns even if they did not provide the actual results for my last concern. They should be given in the revised manuscript. From the other reviewers' comments, I would like to stick to my current rating.

Reviewer 2



Edit: I have read the author response and other reviews. My score remains the same. The paper proposes to frame multi-task learning as multi-objective optimization in the line of Sener and Koltun (NIPS 2018). Importantly, the proposed approach not only finds a single solution on the Pareto frontier, but a diverse set of different solutions that trade-off with different trade-offs. This is achieved by decomposing the MTL problem into K subproblems, each of which is defined by a unit preference vector and constrained to be in a subregion in the objective space. Overall, I found the approach well motivated and the paper well written. I am not aware of prior work that enables the user to trade-off the performance of different tasks for multi-task learning and believe that this may be of practical impact. It is also interesting to see that Pareto MTL is also able to find a good solution if only strong performance on one of the tasks is desired. I also appreciated the comparison to adaptive weight loss approaches, which should enable different perspectives on multi-objective approaches to MTL. I particularly enjoyed the extensive supplementary material, including the analysis of the importance of finding an initial solution, the analysis of the adaptive weight vectors, and an extension to many tasks. There are a few typos in the paper: "neural" -> "natural" (line 27); "depended" -> "dependant" (line 111); "significant" -> "significantly" (line 213); "MultiFashion-MINST" -> "MultiFashionMNIST" (line 245).

Reviewer 3



Originality: medium. This paper mainly combines another MOO algorithm and MOO-MTL, and improves the results from last year NIPS paper Multi-objective MTL. The technical contribution for MOO and MTL is limited since this paper just borrow the MOO optimization method directly from reference [24] and reference [29]. Nevertheless, I think this paper has a potential impact in MTL community since I do not find any previous paper achieves similar effects, which could guide people get different high-quality MTL results without random trails. Quality: below the average The quality of the paper is below the bar. Some important part is missing and lack of deep analysis. 1. The motivation is not clear. The author fails to explain why multi-objective MTL can only find concentrated solutions. The only explanation of this is in Fig 2, which is very empirical. For Fig.2, the author may consider explain it more in paper instead of in the supplementary materials (what is x-axis and y-axis). Also, for the linear scaling, why the solutions with different weight are also concentrated? The author only refers Boyd's optimization book. Maybe the author should consider adding the chapter and pages or give the explanation. 2. How to choose preference vector u? The author mentioned this a little in algorithm and supplementary materials but not enough. My concern is the sensitivity of the preference vector. What is the relationship between vector u and final distribution of the MTL solutions? Also, if we have many tasks instead of two or three, how to choose the vector u? 3. The experiment is performed on only two tasks and three tasks. What is the algorithm complexity respect to the number of the task? The evenly distributed vector u is large in many tasks scenario and there is no empirical and theoretical support that this method will work in many tasks case. Clarity: below the average The writing has a lot to improve. 1. For equation 3, the author only gives the reference, but this is a very important part of the paper and should be introduced with more details. the author should at least add one or two sentences to explain the intuition to make the logic flow more clear. 2. For equation 5-7, the author should make the optimized variable clear. For every objective equation, the author should add the subscript below the max and min notation. Also, eq 16 does not have the subscript for max. 3. Check the subscript more carefully. For example, In Eq 15, I guess the LHS should not have the subscript i? Similar situation is in Eq 18. Significance: above the average The results and the improvement over MOO-MTL are significant and may have an important impact especially in the MTL community. The main reason I reduce the significance to above the average is that the writing is not clear and the logic flow is not rigorous. The supplementary materials do not have theoretical support and the empirical support is not strong enough to convince me the method is robust and applicable in more complicated cases.

[Author Response · NeurIPS 2019]

We sincerely thank all reviewers for their helpful and constructive feedback.

**Response to Reviewer 1:**

-**Pareto solutions for less relevant tasks:** 1) Less relevant tasks often compete with one another, and could lead to
worse performance as noticed in [decaNLP, McCann2018]; 2) Our experiments on Multi-Fashion+MNIST (Fig.4(c),
two less relevant tasks) show that Pareto MTL can still provide a set of widely-distributed Pareto solutions; 3) The
solution with a strong preference on one task (e.g., (1,0)) can achieve the best performance on it, and the other one
can be treated as an auxiliary task; 4) Some recently proposed works on learning tasks relation (e.g., [Ma2018, KDD])
would be useful for Pareto MTL to deal with less relevant tasks. We will discuss it in the revision.

-**Unclear sentences:** 1) L87-88: What we claim is that all those methods try to balance different tasks by adapting the
weights and do not have a systematic approach to incorporate preference information (more details in the supplement:
adaptive weight vectors); 2) L173-175: As discussed in [30], the projection approach (8) is an n-D constrained
optimization problem. It is inefficient to solve it directly, especially for a DNN with millions of parameters. Our
approach reformulates it as an unconstrained problem (to reduce the value of all activated constraints) which can then
be efficiently solved by a gradient-based method; 3) We will rewrite these sentences to make them clear in the revision.

-**Preference vector:** They are evenly distributed on the first quadrant of a unit circle. Five preference vectors for two
tasks are: $\{(cos(\frac{k\pi}{8}), sin(\frac{k\pi}{8}))|k = 0, 1, ..., 4\}$. See response to R4 for random vectors. We will add it in the revision.

-**Better performance:** Short analysis: 1) Using the same argument in [7, 13], ParetoMTL as an adaptive weight method
can outperform fixed weight method; 2) Using the same argument in [12], treating MTL as MOO can obtain better
performance than heuristic-based adaptive weight approach; 3) Compared with MOO-MTL[12], Pareto MTL can find
Pareto solutions with different trade-offs by decomposing the MTL; 4) We will add a detailed analysis in the revision.

-**Advanced networks:** 1) Following the setting in [12], we used LeNet as the basic network for MNIST-like datasets; 2)
Our preliminary results show that a more powerful network can provide better results for all experiments while the
conclusions still hold. We will report these results in the revision.

**Response to Reviewer 2:** We have fixed the typos and carefully proofread the whole paper. We agree that testing
Pareto MTL on tasks from different modalities is important. We will add a comparison and discussion in the revision.

**Response to Reviewer 4:**

-**Why other methods fail:** 1) As shown in [ConvexOptBook, Chapter 4.7.4 Scalarization, Boyd2004], linear scalariza-
tion cannot find the concave part of the Pareto front. We will discuss it in the revision; 2) MOO-MTL tries to balance
different tasks during the optimization process (see the adaptive weight analysis in the supplement), so its solutions are
most likely in the middle of the Pareto front; 3) Our proposed Pareto MTL decomposes a given MTL and guides search
to different sub-regions for obtaining well-distributed solutions; 4) We will move the introduction of the toy example
from the supplement to the main paper and add analysis on the concentrated behaviors.

-**Preference vector:** See response to R1 for vector generation. The
final distribution of the solutions depends on both the preference
vectors and the shape of the Pareto front. Fig.1 shows the effect of
different random prefs. Utilizing prior information/learning-based
method is an important extension to find better-distributed solutions.

-**Many tasks:** 1) Combining many tasks is an important problem
in MTL. As discussed in [decaNLP, McCann2018], simply com-
bining many (and less relevant) tasks might deteriorate their perfor-
mance; 2) Pareto MTL needs to solve an ($n\_\mathbf{tasks} + n\_\mathbf{preferences}$)-
dimensional constrained opt problem to find the descent direction at
each iteration, and solve $n\_\mathbf{preferences}$ subproblems in total, which
would be very time-consuming; 3) However, generating a large num-
ber of Pareto solutions would be useless (think about an MTL prac-
titioner needs to choose 1 among 50 Pareto solutions balancing 15
tasks); 4) For many tasks case, we can still apply Pareto MTL by
a) finding representative solutions with preferred trade-offs and b)
tasks/objectives reduction method from MTL/MOO community.

Figure 1: PMTL w/ different sets of rand prefs can consistently generate well-distributed sols, but too close prefs might lead to worse performance. Discussion will be added in the revision.

-**More accurate equations:** 1) Equation 3 intuition: for finding a descent direction , we either have: a) $\nabla\mathcal{L}_i(\theta_t)^T d_t \leq$
$0, i = 1, ..., m$, which means $d_t$ is a valid descent direction for all tasks or b) no valid descent direction can be found,
the current solution is a Pareto critical point. We will add a paragraph to explain this intuition; 2) We have also fixed the
equations you pointed out and carefully proofread the whole paper to make all equation clear to be understood.

[Meta-Review · NeurIPS 2019]

The paper is extending the recent (NeurIPS 2018) work, which poses multi-task learning as multi-objective optimization. Although the submission is somewhat incremental, it is significant. Finding an arbitrary point on a Pareto efficiency curve is a significant limitation, and practitioners would rather find the entire Pareto efficiency curve. The submission overcomes this limitation. Moreover, empirical results support the claim and show the significance of the method. In the meantime, the submission can still be improved significantly. I strongly recommend the following actions to the authors until the camera-ready deadline: 1 - Improve the presentation of the paper. Some of the reviewers found the paper hard to read. I agree with the concern; although, I find the required edits minor and easy to do before the camera-ready deadline. 2 - Discuss the scalability of the method in terms of both memory and computation. I think the proposed method would have a hard time scaling to a large set of tasks. For example, [12] uses multi-label classification as a multi-task problem with 40 tasks. I am curious how many regions would you need to efficiently cover the Pareto efficiency curve of a problem with 40 tasks? Additional experiments and/or a discussion would be useful.